# Cost-Effectiveness of the Pneumococcal Vaccine in the Adult Population: A Systematic Review

**DOI:** 10.3390/healthcare12232490

**Published:** 2024-12-09

**Authors:** Nam Xuan Vo, Huong Lai Pham, Uyen My Bui, Han Tue Ho, Tien Thuy Bui

**Affiliations:** 1Faculty of Pharmacy, Ton Duc Thang University, Ho Chi Minh City 700000, Vietnam; h1900276@student.tdtu.edu.vn (H.L.P.); h1900341@student.tdtu.edu.vn (U.M.B.); h2100032@student.tdtu.edu.vn (H.T.H.); 2Faculty of Pharmacy, Le Van Thinh Hospital, Ho Chi Minh City 700000, Vietnam; bttien.ths.tcqld23@ump.edu.vn

**Keywords:** pneumococcal vaccine, adult, cost-effectiveness, systematic review, higher-valency vaccine, lower-valency vaccine

## Abstract

**Objectives:** Pneumococcal disease (PD), caused by *S. pneumoniae*, is a serious global health issue, primarily for adults over 65, due to its high mortality and morbidity rates. Recently, broader-serotype vaccines have been introduced to cope with tremendous hospital costs and decreasing quality of life. Our study aims to systematically review the cost-effectiveness of current PCVs (pneumococcal conjugate vaccines) and PPVs (pneumococcal polysaccharide vaccine) from 2018 to April 2024. **Methods:** Articles were identified through PubMed, Embase, and Cochrane. Key outcomes include an improved incremental cost-effectiveness ratio (ICER) and quality-adjusted life-years (QALY), with the article’s quality assessed via the Consolidated Health Economic Evaluation Reporting Standards 2022 (CHEERS 2022). In total, 23 studies were included, with 22 studies of high quality and 1 of moderate quality. **Results:** These articles showed that PCV20 was the most cost-effective option compared with other vaccines, including PPV23, PCV13, PCV15, and PCV15/PPV23, for both young and older adults, regardless of risk factors. PCV20, when used alone, saved greater costs than PCV20, followed by PPV23. **Conclusions:** For countries applying lower-valency vaccines, switching to PCV20 as a single regimen would be the most beneficial for averting pneumococcal cases and reducing costs in adults aged 18–64 and over 65.

## 1. Introduction

Pneumococcal disease (PD), caused by *Streptococcus pneumoniae*, is a significant contributor to vaccine-preventable illnesses and fatalities worldwide [1,2]. Clinically, PD is categorized into invasive pneumococcal disease (IPD) and non-invasive diseases [3], in which pneumococcal pneumonia (as the non-invasive case) is deemed the most frequent manifestation of PD in adults [3,4,5]. In addition, the elderly, particularly those over 65 years of age, are also at substantial risk of contracting PD [6,7]. Furthermore, adults with immunocompromising conditions or chronic comorbidities face a heightened risk of developing IPD and experience higher mortality rates compared to healthy persons [1,8,9].

The global burden of PD in adults is predominantly due to the high incidence of community-acquired pneumonia (CAP), with a varying distribution across regions [10]. Even though CAP is broader than pneumococcal pneumonia (PP) in terms of being a causative pathogen, it is still used as a tool to reflect the burden of overall PD since *Streptococcus pneumoniae* contributes as the most prevalent pathogen, present in up to 60% of CAP episodes [11,12]. In the United States, the rate of CAP infections fluctuates between 24.8 per 10,000 and 106 per 10,000 among those aged 18–64, though it was much higher in those over 65, with rates of approximately 63.0 per 10,000 person-years for ages 65–79, and reaching 164.3 per 10,000 person-years for those over 80 [13]. The annual rate of mortality for CAP is 24.9% [14].

The economic impact of PD is significant. In 2017, the entire cost of PD in adults over 19 in the USA was projected at USD 1.86 billion, with USD 1.8 billion going toward direct medical costs [15]. Regarding pneumonia, EUR 10–12 billion is spent annually in Europe [16]. Notably, hospitalization is the primary cause of the enormous expense [11,17]. Non-invasive cases (known as non-bacteremic pneumococcal pneumonia—NBPP—in clinical practice) are the most prevalent, accounting for 3/4 inpatients diagnosed with PP [4,5]. In Australia, in 2011–2012, hospitalized PP patients accounted for AUD 50 million, while the IPD case cost was about AUD 1.2 million in total [16]. The number of hospitalized PP cases in the United States is expected to nearly double between 2004 and 2040 as the elderly population grows, resulting in an additional USD 2.5 billion in annual costs [18]. In addition, the accumulated indirect cost owing to productivity loss was significant. In Poland, PP accounted for 89.8% of absenteeism (sickness absence) in 2017, which was equivalent to 49.4 million Polish złoty (PLN) of indirect costs [19]. In terms of IPD, again, it is still pneumonia that remains the most frequent diagnosis, with bacteremic PP affecting 80–90% of total cases [20]. In countries with advanced economies, the incidence of IPD is between 10 and 50 per 100,000 persons per year [1].

Vaccination has proven to be the most effective defense against complex PD. Over the last few decades, two vaccines for adults have been recommended: a 13-valent pneumococcal conjugate (PCV13) and a 23-valent polysaccharide (PPV23) [6]. PCV13 plays a crucial role in protecting adults from vaccine serotype community-acquired pneumonia (VT-CAP) to compensate for PPV23’s drawbacks [21]. The term “vaccine type” refers to the contributing serotypes within the specific vaccine coverage. For instance, VT-IPD in PCV13 means the IPD cases caused by the serotype covered in PCV13, and VT-CAP in PPV23 illustrates the episodes caused by the serotype covered in PPV23. Other than direct protection through vaccination, widespread uptake of PCVs (primarily PCV13) in infants has greatly diminished vaccine-type PD cases in adult populations (including the unvaccinated) via herd immunity or indirect effects [21,22]. This trend has been noticed in several high-income countries that have incorporated PCVs into pediatric national immunization programs (NIPs), such as the USA [22], UK [23], Germany [24], and Italy [6]. Nonetheless, the burden of PD remains, with a surge in non-vaccine serotypes or serotype replacement [25,26,27,28], notably in cases of IPD throughout many countries. According to Hanquet et al., a review of non-vaccine type changes related to IPD in ten European countries employing PCV10 or PCV13 in pediatric NIPs revealed an increase in non-PCV13 serotype cases [29]. To tackle this issue, higher-valency vaccines with broader serotype coverage, such as a 15-valent pneumococcal conjugate vaccine (PCV15) and a 20-valent pneumococcal conjugate vaccine (PCV20), were approved in the USA in 2021 for adults over 18 years old [30]. PCV15 offers two additional serotypes, while PCV20 adds seven more serotypes in addition to those covered by PCV13 [31]. Unlike PCV13, which has well-established vaccine effectiveness proven through various trials and observational studies, PCV15 and PCV20 were licensed based on immunogenicity trials, which indicated non-inferiority compared to PCV13 [32,33]. As a result, there is a lack of direct efficacy data on higher-valency regimens. Therefore, cost-effectiveness analysis is the foundation for decision-makers to ascertain the value of these vaccinations while being aware of the necessary cost. Given the complexity of vaccination strategies, it is critical to analyze the cost-effectiveness of the latest pneumococcal vaccines so that stakeholders and policymakers can make informed decisions on resource allocation based on their country’s circumstances. Therefore, we aim to systematically review adult cost-effectiveness studies to highlight the economic benefits of the current programs (PCV13 and PPV23) and higher-valency vaccinations (PCV15 and PCV20).

## 2. Method

### 2.1. Searching Strategy

The systematic review followed the PRISMA 2020 checklist [34]. The time length for identifying articles related to pneumococcal vaccines in adults was limited from 2018 to 2024. Studies were selected from the last 5 years (2018–2023), and the final update was conducted in 2024 (one study was added) to update the latest knowledge developed based on changes in science and technology, changes in socio-economic conditions, and policies in different countries. Data were searched in English from the following electronic databases PubMed, Embase, and Cochrane, with the most recent article being accessed in April 2024. To identify relevant publications, we also used key terms such as “pneumococcal disease”, “adults”, “PCV”, “PPV”, “vaccination”, “economic evaluation”, or “cost-effectiveness analysis”. The indexing process adhered to the following criteria: (pneumococcal) AND ((cost-effectiveness) OR (economic evaluation)) AND ((PCV) OR (PPV) OR (vaccination)) AND ((adult) OR (elderly)). The research has been registered in the Open Science Framework (https://osf.io/, https://osf.io/m29y7 (accessed on 5 November 2024)).

### 2.2. Selection Process and Criteria for Research Selection

The articles were evaluated based on the following inclusion criteria: (1) The scope of economic analysis covers cost-effectiveness analysis (CEA), cost–benefit analysis (CBA), or cost–utility analysis (CUA); (2) the examination focuses on an adult population or specific subgroups over the age of 18 and no limitations are placed on the participant count and risk classification; (3) a comparison has been made among specific pneumococcal vaccines; (4) the study presents data regarding health outcomes such as quality-adjusted life-years (QALY), the incremental cost-effectiveness ratio (ICER), or life-years (LY); (5) there is a definitive conclusion regarding the cost-effectiveness of each intervention if mentioned; (6) a full-text review is available; (7) the publication is in English.

The exclusion criteria were as follows: (1) The scope of economic analysis does not encompass the study; (2) the participants are under 18 years of age or those under 18 years old are included in the subgroups being compared; (3) there is an unclear specification of the comparison between pneumococcal vaccines in adults; (4) the research does not reach a definite conclusion about the cost-effectiveness results among particular types of pneumococcal vaccines; (5) there is a lack of health outcome data (ambiguous/indeterminate health outcome data); (6) a full-text review is unavailable; (7) the publication is in any language other than English.

Two independent reviewers initially checked the title and abstract for each article based on the inclusion conditions. Articles meeting these criteria were read as the full text, and the relevant information was extracted. Any obtained data required a consensus between the two reviewers throughout the selection process.

### 2.3. Data Extraction

The following information was extracted: the first author, publication year, country, analysis type, intervention, study design, clinical outcomes, method approach, population, risk group, time horizon, discount rate, health outcomes, sensitivity analysis, currency, perspective, vaccine coverage, and funding source. We estimate the primary outcome using the ICER (incremental cost-effectiveness ratio), which is the ratio of the difference in costs to the difference in effectiveness between two interventions [35]. The ICER of each economic study was grounded in the base-case analysis. Also, the primary metrics in healthcare evaluation are QALY (quality-adjusted life-years) or LY (life-years). The formula is stated as follows:$$ICERs=\frac{Cost\;A-Cost\;B}{Effectiveness\;A-Effectiveness\;B}$$

The ICERs will then be compared to a monetary threshold called WTP (willingness to pay) to check the affordability of that intervention. WTP refers to the maximum amount of money an individual or group is willing to spend or pay to obtain a particular good or service or achieve a specific outcome or benefit [36]. An intervention is considered “cost-effective” if the ICERs fall within the WTP range; however, if they exceed the WTP, it is regarded as “not cost-effective” [36]. The most common unit of WTP is based on the cost per QALY [36]. Another benchmark is based on the national annual gross domestic product (GDP) per capita, following the World Health Organization’s Choosing Interventions that are Cost-Effective (WHO-CHOICE) recommendation [37]. According to this guideline, if the ICERs are within three times the GDP per capita, the intervention is considered cost-effective; if the ICERs are less than one GDP per capita, it is deemed highly cost-effective [37]. There are also three types of cost: vaccine, direct, and indirect. The direct cost includes medical and non-medical costs, while the indirect cost comprises productivity loss.

Sensitivity analyses from each study were also examined to determine the robustness of the ICERs in the base case. The most commonly used sensitivity analyses include deterministic sensitivity analysis (DSA) and probabilistic sensitivity analysis (PSA). In DSA, the most influential inputs affecting ICERs are identified. In PSA, the results are often presented as scatter plots or cost-effectiveness planes to comprehensively understand the relationship between costs and effectiveness across different interventions.

The population under study primarily consisted of adults likely to have underlying conditions. These adults can be categorized as either immunocompetent or immunocompromised. The immunocompetent target group encompasses those with a low risk (healthy people) and moderate risk (individuals with chronic medical conditions or unhealthy lifestyle behavior such as alcoholism, smoking, heart disease, chronic liver disease, chronic lung disease, or diabetes mellitus) [38]. Immunocompromised groups include those who suffer from immune deficiency, HIV infection, chronic renal failure, splenic dysfunction, sickle cell disease, and illnesses requiring treatment with immunosuppressive drugs (Hodgkin disease, leukemia, organ transplant…) [38].

### 2.4. Quality Assessment

The 28-item Consolidated Health Economic Evaluation Reporting Standards 2022 (CHEERS 2022) checklist was used to review and summarize the methodological quality of all critically included articles [39]. The scoring method of the CHEERS 2022 checklist does not follow a fixed formula but is flexible depending on each specific study. Instead, the assessment focuses on considering the report’s response level to each criterion in the checklist. Researchers often use a scale to assess, for example, 0—not responsive, 0.5—partially responsive, and 1—fully responsive. After assessing each criterion, the overall score of the report will be calculated, which may consider adding weight to essential criteria. The scoring method of the CHEERS 2022 checklist combines both objectivity and subjectivity. Although there is a common standard framework, assessing a report’s response level to each criterion still requires the reviewer’s judgment. Experts often use scales to quantify the level of satisfaction, but determining the weight for each criterion can vary depending on the research goals and individual perspectives. To ensure objectivity, many studies use independent assessments by two or more people, then compare and unify the results. In this research, two investigators independently screened the literature, and then summaries of the results were discussed. The article’s assessment scores consist of three values: 0, 0.5, and 1, with “0” if the information was found to be irrelevant, “0.5” for partially given responses, and “1” indicating a completely fulfilled response. Next, by assigning each study a total score, we classified their reporting quality into three categories: high quality (total scores above 21), moderate quality (total scores between 14 and 21), and poor quality (total scores below 14).

## 3. Results

### 3.1. Selection Process

Firstly, the economic evaluation articles of PCV vaccines by searching various online databases such as PubMed, Embase, and Cochrane, starting from January to April 2024. Any articles from 2018 to date were within our range of search. In total, 838 articles were identified through searching, with 280 articles from PubMed, 542 from Embase, and 16 from Cochrane. Notably, the last publication was retrieved on 30 April 2024. After removing 73 duplicates, 765 articles remained. The title, abstract, and full-text screening were applied based on the exclusion and inclusion criteria. As a result, 740 articles were excluded, potentially resulting in only 25 qualified articles. Upon examining the full texts, we determined that two articles were unsuitable. Although Ojal et al. calculated the ICER for pediatric and adult patients, the research failed to provide conclusions specific to the adult population [40]. On the other hand, Willem et al.’s study lacked data related to economic health outcomes, including QALY gain, incremental cost, and specific ICERs [4]. Hence, these two articles were disqualified. Overall, 23 investigations were qualified for synthesis. The selection process was shown in Figure 1 below:

### 3.2. Characteristics of Selected Articles

The overall information of the chosen evaluation is demonstrated in Table 1. Cost-effectiveness analysis (CEA) was the most frequently used analysis type, appearing in 16 out of 23 articles (69.56%) [5,41,42,43,44,45,46,47,48,49,50,51,52,53,54,55]. Then, CUA was employed in 6/23 studies (26.1%) [56,57,58,59,60,61]. Notably, one article utilized both CEA and CUA [62]. A large amount of this research was carried out in developed countries, with more than half (13/23 articles) originating from Europe, namely Portugal [62], Sweden [44], Denmark [57], England [47], Norway [56,58], Belgium [59], Spain [60], Italy [55,61], the Netherlands [52,54], and Greece [5]. In addition, 4 of the 23 analyses were undertaken in the United States [46,48,51,53]. Asia also contributed to a partial proportion, with 4/23 articles coming from Japan [41], Thailand [42], China [43], and South Korea [45]. The rest were from Canada [49] and Argentina [50], which lie in North and South America, respectively.

Regarding interventions, 12/23 studies discussed the long-circulated pneumococcal vaccines or lower-valency immunization, which included PCV13 and PPV23 with or without sequential administration [41,42,43,44,45,46,49,50,52,53,58,62]. In parallel, higher-valency pneumococcal vaccines, such as PCV15 and PCV20, were examined in 11/23 articles [5,47,48,51,54,55,56,57,59,60,61]. All of the interventions were assumed to compare one dose of PCV or PPV. Moreover, 16 out of 23 (69.56%) economic evaluations focused on older adults, particularly those ≥65 years (11/16) [42,44,45,46,49,50,51,53,54,58,61]. In addition, two studies discussed those aged ≥60 years (02/16 analysis) [52,60] and three studies examined the 50–64 years age group (3/16 articles) [41,43,53]. The remaining studies were concerning the groups starting from 18 years old (7/23 articles, 30.44%) [5,47,55,56,57,59,62]. In terms of medical conditions, most of the studies covered both immunocompetent and non-immunocompetent individuals, as stated in 17/23 articles (73.9%). In contrast, only two research focused solely on immunocompetent patients [43,48], and four studies did not specify the medical status of their subjects [44,52,54,58].

The Markov model was predominantly employed, with around 78% of the studies (18/23) applying it [5,42,43,45,46,47,48,49,50,51,53,55,56,57,59,60,61,62]. Most of the studies considered the healthcare perspective, which was recorded in 14/23 articles, accounting for nearly 61% [43,44,45,46,47,48,49,51,52,56,58,59,60,61]. Only one article by Igarashi et al. in Japan applied more than one perspective, particularly the payer and society [41], while Giglio et al. did not mention this [50]. The others were performed from the payer’s or society’s perspective, as seen in the majority of developed countries such as the United States, Japan, and European nations like Denmark, Portugal, the Netherlands, Italy, and Greece, with the payer’s perspective recorded in 2/23 articles [53,55] and the societal perspective in 5/23 articles [42,54,55,57,62].

The time horizon ranged from 5 years to 100 years (lifetime), with 15/23 articles (69.6%) applying 100 years on average [5,41,42,43,45,46,47,48,51,55,56,57,59,61,62]. The discount rate varied from 1.50% to 5%, with the most widely used level being 3% in 14 out of 23 publications (60.9%) [42,44,45,46,48,50,51,53,55,56,59,60,61,62]. The primary sponsors were biopharmaceutical companies, with Pfizer leading at 47.8%, supporting 11 out of 23 studies [5,41,45,47,49,56,57,59,60,61,62], and Merck adding to one research study [55]. The National Institute of Allergy and Infectious Diseases was the second-largest sponsor, financing 3 of the 23 articles [48,51,53]. Additional funding sources came from national and international organizations like Mahidol University, the National Technology Key Research Program of China, the Belgian Health Care Knowledge Center, the National Institutes of Health, I-Move+, the National Institute for Health Research Health Protection Research Unit, and the Netherlands Ministry of Health, Welfare, and Sport. Notably, 2 of the 23 articles, or 8.7%, received no funding source [44,58].

**Table 1 healthcare-12-02490-t001:** Characteristic features of chosen economic evaluations.

No.	Author, Year, Country	EE Type	Intervention	Age (Years)	Risk Profile	VC	Clinical Outcome	Model	Time Horizon	Discount Rate	Currency	Perspective	Funding	Health Outcome	SA
1	Igarashi et al., 2021, Japan [41]	CEA	PCV13 vs. No vaccination PPV23 vs. No vaccination PCV13 vs. PPV23	60–64	Immunocompetent Immunocompromised	-	IPD NBPP: in/out	Natural history model	Lifetime	2%	2021 JPY	Payer, Society	Pfizer	QALY, LY, ICER	DSA, PSA
2	Gouveia et al., 2019, Portugal [62]	CEA, CUA	PCV13 vs. No vaccination PCV13 vs. PPV23	≥18	Immunocompetent Immunocompromised	100%	IPD ACP: in/out	Markov	Lifetime	3%	2014 EUR	Society	Pfizer	QALY, LY, ICER	DSA, PSA
3	Ngamprasertchai et al., 2023, Thailand [42]	CEA	PCV13 vs. No vaccination PPV23 vs. No vaccination	≥65	Immunocompetent Immunocompromised	-	IPD NBPP	Markov	Lifetime	3%	2021 USD	Society	Mahidol University	QALY, LY, ICER	DSA, PSA
4	Sun et al., 2021, China [43]	CEA	PPV23 vs. No vaccination	≥60	Immunocompetent: diabetic	-	IPD CAP: in/out	Markov	Lifetime	5%	2013 USD	Healthcare	NTKRPC	QALY, ICER	DSA
5	Wolff et al., 2020, Sweden [44]	CEA	PPV23 vs. No vaccination PCV13 vs. No vaccination	≥65 ≥75	-	75%	IPD pCAP: in/out	Decision tree	5 years	3%	2020 EUR	Healthcare	None	QALY, ICER	DSA
6	Malene B et al., 2023, Norway [56]	CUA	PCV20 vs. PPV23	18–99	Immunocompetent Immunocompromised	75%	IPD NBPP: in/out	Markov	Lifetime	0–39 years: 4% 40–74 years: 3% >75 years: 2%	2022 EUR	Healthcare	Pfizer	QALY, ICER	DSA
7	Choi et al., 2018, South Korea [45]	CEA	PCV13/PPV23 vs. PPV23 PCV13 vs. PPV23	≥65	Immunocompetent immunocompromised	100%	IPD NBPP: in	Markov	Lifetime	3%	2016 USD	Healthcare	Pfizer	QALY, ICER	DSA
8	Olsen et al., 2022, Denmark [57]	CUA	PCV20 vs. PPV23 PCV20/PPV23 vs. PPV23	≥18 ≥65	Immunocompetent Immunocompromised	73%	IPD NBP: in/out	Markov	Lifetime	0–35 years: 3.5% 36–70 years: 2.5% >70 years: 1.5%	2022 EUR	Society	Pfizer	QALY, ICER	DSA, PSA
9	Wateska et al., 2020, USA [46]	CEA	PPV23 vs. No vac	≥65	Immunocompetent Immunocompromised	-	IPD NBPP: in/out	Markov	Lifetime	3%	2014 USD	Healthcare	NIH	QALY, ICER	DSA, PSA
10	Mendes et al., 2022, England [47]	CEA	PCV20 vs. PCV15/PPV23 PCV20 vs.PCV20/PPV23 PCV20 vs. PPV23	≥18	Immunocompetent immunocompromised	-	IPD CAP: in	Markov	Lifetime	3.5%	2019 EUR	Healthcare	Pfizer	LY, QALY	PSA
11	Nymark et al., 2022, Norway [58]	CUA	PPV23 vs. No vaccination	≥65 ≥75	-	75%	IPD pCAP	Decision tree	5 years	4%	2022 EUR	Healthcare	None	QALY	DSA
12	Marbaix et al., 2023, Belgium [59]	CUA	PCV20 vs. No vaccination PCV20 vs. PCV15/PPV23	≥18	Immunocompetent Immunocompromised	15–18%	IPD NBP: in/out	Markov	Lifetime	3%	2023 EUR	Healthcare	Pfizer	QALY, ICER	DSA, PSA
13	Cantarero et al., 2023, Spain [60]	CUA	PCV20 vs. PCV15/PPV23	≥60	Immunocompetent Immunocompromised	17%	IPD NBP: in/out	Markov	10 years	3%	2018 EUR	Healthcare	Pfizer	QALY, LY, ICER	DSA, PSA
14	Polistena et al., 2022, Italy [61]	CUA	PCV20 vs. PCV13 PCV20 vs. PCV15	65–74	Immunocompetent Immunocompromised	-	IPD NBP: in/out	Markov	Lifetime	3%	2022 EUR	Healthcare	Pfizer	QALY	DSA, PSA
15	Smith et al., 2021, USA [48]	CEA	PCV20 vs. PPV23 PCV20/PPV23 vs. PPV23	≥65	Immunocompetent	-	IPD NBPP: in	Markov	Lifetime	3%	2017 USD	Healthcare	NIAID	QALY	DSA, PSA
16	Smith et al., 2022, USA [51]	CEA	PCV20 vs. No vac PCV15/PPV23 vs. No vac	≥65	Immunocompetent Immunocompromised	-	IPD NBPP: in/out	Markov	Lifetime	3%	2017 USD	Healthcare	NIAID	QALY, ICER	DSA, PSA
17	Thorrington et al., 2018, Netherlands [52]	CEA	PPV23 vs. No vac PCV13 vs. No vac	≥60	-	50%	IPD CAP: in	Static model	10 years	4% cost, 1.5% health	2018 EUR	Healthcare	I-Move+, NIHR HPRU	QALY, ICER	-
18	Wateska et al., 2019, USA [53]	CEA	PPV23 vs. PCV13/PPV23 PPV23 vs. PCV13 PPV23 vs. Expanding PPV23 uptake	50–64	Immunocompetent Immunocompromised	-	IPD NBPP	Markov	50 years	3%	2015 USD	Payer	NIAID	QALY	DSA, PSA
19	Boer et al., 2024, Netherlands [54]	CEA	PCV20 vs. No vac x3 PPV23 vs. No vac PCV15 vs. No vac	≥65	-	70%	IPD NBPP: in	Static model	15 years	4%	2021 EUR	Society	Netherlands Ministry of Health, Welfare, and Sport	QALY, ICER	DSA, PSA
20	Restivo et al., 2023, Italy [55]	CEA	PCV15/PPV23 vs. PCV13/PPV23 PCV15/PPV23 vs. PCV20/PPV23 PCV15/PPV23 vs. PCV20 PCV20/PPV23 vs. No vac	≥18	Immunocompetent Immunocompromised	25–65%	IPD NBP: in/out	Markov	Lifetime	3%	2021 EUR	Society	Merck	QALY, LY	DSA, PSA
21	Gourzoulidis et al., 2023, Greece [5]	CEA	PCV20 vs. PCV15 PCV20 vs. PCV15/PPV23	≥18	Immunocompetent Immunocompromised	-	IPD NBP: in/out	Markov	Lifetime	3.5%	2022 EUR	Payer	Pfizer	QALY	DSA, PSA
22	Atwood et al., 2018, Canada [49]	CEA	PCV13/PPV23 vs. PPV23	≥65	Immunocompetent Immunocompromised	-	IPD ACP: in/out	Markov	5 years	5%	2014 CAN	Healthcare	Pfizer	QALY	DSA, PSA
23	Giglio et al., 2022, Argentina [50]	CEA	PCV13/PPV23 vs. PPV23	≥65	Immunocompetent Immunocompromised	85%	IPD: bacteremia Pneumonia: in/out	Markov	10 years	3%	2020 USD	-	Pfizer	LY, ICER	DSA, PSA

Abbreviation: VC: vaccine coverage; NBP: non-bacteremic pneumonia; NBPP: non-bacteremic pneumococcal pneumonia; CAP: all-cause community-acquired pneumonia pCAP: pneumococcal community-acquired pneumonia; in/out: inpatient and outpatient; NTKRPC: National Technology Key Research Program of China; NIH: National Institute of Health; NIAID: National Institute of Allergy and Infectious Diseases; NIHR HPRU: National Institute for Health Research Health Protection Research Unit; No vac: no vaccination; PCV13/PPV23: PCV13 followed by PPV23, PCV15/PPV23: PCV15 followed by PPV23; PCV20/PPV23: PCV20 followed by PPV23, x3 PPV23: three doses of PPV23; SA: sensitivity analysis DSA: deterministic sensitivity analysis; PSA: probabilistic sensitivity analysis.

### 3.3. Risk of Bias

The quality outcomes of our chosen economic studies are shown in the Appendix A. After assessing the quality of the 23 studies, 22 were of high quality (22 points or more), and 1 was of moderate quality. For the 22 high-quality studies, 1 study scored 24/28 points (4.55%) [5], 7 studies scored 23.5/28 points (31.82%) [41,42,52,54,56,58,59] and 8 studies scored 23/28 points (36.36%) [43,45,47,49,55,57,60,62], 3 studies scored 22.5/28 points (13.64%) [44,48,61], and 3 studies scored the lowest score of 22/28 points (13.64%) [46,51,53]. Only one article was rated as moderate quality, with a score of 15.5/28 due to insufficient methodological detail [50]. Most studies (22/23) were rated as high quality, with a score of 22/28 or higher, which indicates that the selected studies had a transparent methodology, complete data, and reliable results. None of the publications lacked information on four specific criteria: Characterization of heterogeneity—Item 18: Studies did not address differences between study populations; Characterization of distribution effects—Item 19: Studies did not analyze the impact of the intervention on different population groups; Patient access and participation—Item 21: Studies did not describe how to collect patient opinions and involvement during the study; and Impact of patient participation—Item 25: Studies did not assess the impact of patient participation on study outcomes.

### 3.4. Incremental Health Outcome of Pneumococcal Vaccines

#### 3.4.1. Lower-Valency Pneumococcal Vaccine

Out of 12/23 articles discussing lower-valency pneumococcal vaccines, the majority compared between single-use vaccination only (7/12) [41,42,43,44,52,58,62], while the rest debated sequence use (5/12) [45,46,49,50,53]. The results regarding the incremental health outcomes and sensitivity analysis are displayed in Table 2a and Table 2b, respectively. Compared to no vaccination, using PCV13 or PPV23 was associated with an additional cost and improved quality-adjusted life-years (QALY) and life-years (LY) in older adults ≥65 years [41,44,52,58,62] in all-risk patients, in immunocompetent patients [43,46], and in the immunocompromised population [42]. Notably, vaccine cost was the primary driver of increased total costs.

Among the four publications concentrating on PCV13 and PPV23, three concluded that a single dose of PCV13 was a more cost-effective option [41,42,62]. Only one article by Thorrington et al. debated that PPV23 was an economical choice over PCV13 in people >60 years old within a WTP of 20,000 EUR/QALY, given the concurrent use of PCV10 in the pediatric population [52].

On the other hand, sequential-use PCV13/PPV23 was estimated to be cost-effective (more costly, extended QALY) compared to single-use PPV23 in all-risk people over 65 in South Korea [45] and Canada [49]. Similarly, in Argentina, PCV13/PPV23 was shown to have a greater LY gain and a dominant ICER, resulting in it being a cost-saving option over PPV23 use alone [50]. When restricted to immunocompetent people (those with or without chronic medical conditions), PPV23 was preferable to a strategy using PCV13, followed by PPV23 in people above age 65, given the WTP of 200,000 USD/QALY gain, according to one article in the USA by Wateska et al., 2023 [46]. In contrast, the research, within the age 50–64, utilizing PCV13/PPV23 led to a negative incremental total cost with extended QALY, resulting in a dominant ICER compared to PPV23 alone [53]. In addition, in over-65 populations with an immunodeficiency condition, two investigations concluded that PCV13/PPV23 was a cost-effective strategy compared to PPV23 [45,49].

**Table 2 healthcare-12-02490-t002:** (a) Incremental cost and health outcomes in lower-valency pneumococcal vaccines. (b) ICERs and sensitivity analysis outcomes of lower-valency pneumococcal vaccines.

**(a)**									
**Country, Currency, Ref**	**Intervention**		**Age**	**Vaccine Cost**	**Medical Cost**	**Indirect Cost**	**Total Cost**	**QALY**	**LY**
**All risk ***									
Japan, 2021, JPY [41]	PCV13 vs. No vaccination PPV23 vs. No vaccination PCV13 vs. PPV23		60–64	10,230 7736 2494	−1598 −853.05 −745.40	1642 1699 −57.28	10,274 8582 1691	0.0076 0.0041 0.0035	0.0087 0.0047 0.0040
Portugal, 2014, EUR [62]	PCV13 vs. No vaccination PCV13 vs. PPV23		≥65	-	-	-	46.65 33.11	0.003 0.003	0.004 0.004
Sweden, 2020, EUR [44]	PPV23 vs. No vaccination		≥65 ≥75	3,297,459 2,252,084	−893,859 −1,041,454	- -	2,526,940 1,333,970	27 45	- -
Norway, 2022, EUR [58]	PPV23 vs. No vaccination		≥65 ≥75	3,085,982 1,771,496	−1,949,287 −1,883,861		1,263,986 14,927	15.91 15.48	- -
Netherlands, 2018, EUR [52]	PPV23 vs. No vaccination		60 65 70	- - -	- - -	- - -	13,144,010 9,845,099 6,403,943	909 1031 1033	- - -
	PCV13 vs. No vaccination		60 65 70	- - -	- - -	- - -	60,792,248 54,040,383 41,029,247	910 1227 1161	- - -
South Korea, 2016, USD [45]	PCV13 vs. PPV23 PCV13/PPV23 vs. PPV23		≥65	- -	- -	- -	- -	- -	- -
Canada, 2014, CAN [49]	PCV13/PPV23 vs. PPV23		≥65	254,300,000	−135,600,000	-	118,700,000	0.0006	1,100,000
Argentina, 2020, USD [50]	PCV13/PPV23 vs. PPV23		≥65	-	-	-	21,667,742	-	716.44
**Immunocompetent**									
Thailand, 2021, USD [42]	PCV13 vs. No vaccination PPV23 vs. No vaccination		≥65	- -	- -	- -	5.67 18.27	0.02 0.01	0.06 0.02
China, 2013, USD [43]	PPV23 vs. No vaccination		≥60	-	-	-	1,962,000	10,321	-
USA, 2014, USD [46]	PPV23 vs. No vaccination PCV13/PPV23, program vs. No vaccination		≥65	- -	- -	- -	67.45 67.87	0.00030 0.00009	- -
USA, 2019, USD [53]	PCV13/PPV23 vs. PPV23		50–64	-	-	-	−0.32	0.00043	-
	PCV13/PPV23		All vs. Chronic	-	-	-	39.16	0.00068	-
**Immunocompromised**									
Thailand, 2021, USD [42]	PCV13 vs. No vaccination PPV23 vs. No vaccination		≥65	- -	- -	- -	12.31 30.98	0.02 0.23	0.73 0.68
South Korea, 2016, USD [45]	PCV13 vs. PPV23 PCV13/PPV23 vs. PPV23		≥65	- -	- -	- -	- -	- -	- -
Canada, 2014, CAN [49]	PCV13/PPV23 vs. PPV23		≥65	149,800,000	−120,300,000	-	29,500,000	0.0009	1,600,000
**(b)**									
**Country, Currency, Ref**	**Intervention**	**Age**	**ICERs**	**The Most Impactful Parameter in DSA**		**PSA**		**Conclusion**	
					**% CE**	**WTP**	**Scatter Plot Distribution**		
**All risk ***									
Japan, 2021, JPY [41]	PCV13 vs. No vaccination PPV23 vs. No vaccination PCV13 vs. PPV23	60–64	1,356,218 2,103,602 483,867	VE in NBP, discounting	98.0% 87.7% 89.3%	5,000,000	100% NE 99% NE 100% NE	PCV13 was more cost-effective than PPV23 or no vaccination	
Portugal, 2014, EUR [62]	PCV13 vs. No vaccination PCV13 vs. PPV23	≥18	17,746 13,146	PCV13 effectiveness	94% 94%	20,000	- -	PCV13 was more cost-effective than PPV23 or no vaccination	
Sweden, 2020, EUR [44]	PPV23 vs. No vaccination	≥65 ≥75	93,578 29,468	Vaccine effectiveness, VT pneumococcal disease	-	50,000	-	PPV23 can be cost-effective at 75 years but not 65 years	
Norway, 2022, EUR [58]	PPV23 vs. No vaccination	≥65 ≥75	79,451 964	Vaccination coverage	-	28,004–84,011	-	PPV23 was cost-effective in the 65- and 75-year-old cohorts	
Netherlands, 2018, EUR [52]	PPV23 vs. No vaccination	60 65 70	14,452 9553 6201	Mortality rate, total cost of program implementation	-	20,000	-	PPV23 was the most cost-effective strategy in the projection of using PCV10 in infants	
	PCV13 vs. No vaccination	60 65 70	66,796 44,028 35,346		-		-		
South Korea, 2016, USD [45]	PCV13 vs. PPV23 PCV13/PPV23 vs. PPV23	≥65	1421 3300	Vaccine effectiveness of PCV13 against NBPP, incidence of NBPP	-	GDP: 27,633	100% NE	PCV13/PPV23 was more cost-effective than PPV23 regardless of co-morbidity	
Canada, 2014, CAN [49]	PCV13/PPV23 vs. PPV23	≥65	35,484	ACP hospitalization cost, PCV13-VE against VT- NBPP	100%	50,000	49% NE	PCV13/PPV23 was more cost-effective than PPV23	
Argentina, 2020, USD [50]	PCV13/PPV23 vs. PPV23	≥65	Dominant **	Percentage of adjustment for PP rate by urine analysis, at-risk pneumonia inpatient cost, pneumonia incidence	-	-	98% SE	PCV13/PPV23 was a cost-saving option	
**Immunocompetent**									
Thailand, 2021, USD [42]	PCV13 vs. No vaccination PPV23 vs. No vaccination PCV13 vs. PPV23	≥65	233.63 1439.25 -	Fatality of NBPP, PCV13 pneumonia efficacy	- - 80%	5003	- Majority NE 80% NE	PCV13 dominated over PPV23	
China, 2013, USD [43]	PPV23 vs. No vaccination	≥60	190.1	VT-effectiveness against CAP, epidemiological data for CAP, administrative costs for PPV23	-	GDP: 14,759	-	PPV23 was cost-effective	
USA, 2014, USD [46]	PPV23 vs. No vaccination PCV13/PPV23, program vs. No vaccination	≥65	226,733 765,018	Robust	30% 5%	200,000	-	PPV23 was more cost-effective than PCV13/PP23	
USA, 2015, USD [53]	PCV13/PPV23 vs. PPV23	50–64	Dominant	PCV13 vaccine price	60.3% 14.8%	50,000 100,000	-	PCV13/PPSV23 was more cost-effective than PPV23, being the least costly in ≥50-year-old people with chronic conditions	
	PCV13/PPV23	All vs. chronic	57,786	PCV13 vaccine price	37% 82.9%	50,000 100,000	-		
**Immunocompromised**									
Thailand, 2021, USD [42]	PCV13 vs. No vaccination PPV23 vs. No vaccination PCV13 vs. PPV23	≥65	627.24 136.13 -	Utility in the elderly, VT-efficacy against pneumonia	- 70% at 250 90% at 4000	5003	- 100% NE Majority NE	PCV13 was more cost-effective than PPV23	
South Korea, 2016, USD [45]	PCV13 vs. PPV23 PCV13/PPV23 vs. PPV23	≥65	1736 3404	PCV13 VE against VT-NBPP Incidence of NBPP	-	GDP: 27,633	100% NE	PCV13/PPV23 was more cost-effective than PPV23 regardless of co-morbidity	
Canada, 2014, CAN [49]	PCV13/PPV23 vs. PPV23	≥65	10,728	PCV13 VE against vaccine-type NBPP	100%	50,000	82% NE	PCV13/PPV23 is incredibly cost-effective in high-risk adults	

Abbreviation: QALY: quality-adjusted life-year; LY: life-years; PCV13/PPV23, program: PCV13 followed by PPV23 plus expand vaccine uptake; PCV13/PPV23: PCV13 followed by PPV23. All risk *: including immunocompetent and immunocompromised groups. PCV13/PPV23, program: PCV13 followed by PPV23 plus expanded vaccine uptake; PCV13/PPV23: PCV13 followed by PPV23; NBPP: non-bacteremic pneumococcal pneumonia; NBP: non-bacteremic pneumonia; PP: pneumococcal pneumonia ACP: all-cause pneumonia; CAP: community-acquired-pneumonia VT: vaccine type; VE: vaccine effectiveness; SE: South East, NE: North East, DSA: deterministic sensitivity analysis; PSA: probabilistic sensitivity analysis; ICERs: incremental cost-effectiveness ratio; WTP: willingness to pay. All risk *: including immunocompetent and immunocompromised groups. Dominant **: The unit of ICER is the cost per life-year.

The results from the DSA indicated that the parameters that affected the ICER the most were vaccine effectiveness, which was reported in 6/12 studies [41,42,44,45,49,62], and the incidence rate of pneumonia, recorded in 4/12 articles [42,44,45,50] (Table 3b). The majority of the simulation was focused on the North East quadrant when comparing PCV13 and PPV23, which proved that PCV13 required a higher cost but also improved quality of life [41,42] in the three risk groups. Within the given WTP, PCV13 was more cost-effective than PPV23.

#### 3.4.2. Higher-Valency Pneumococcal Vaccine

In 10 of the 11 selected articles on the most recent vaccine, PCV20 as a single immunization was identified as the dominant strategy compared to other interventions for younger and older adults over 65, irrespective of their risk profiles. The comprehensive results of the cost-effectiveness data and sensitivity analysis using the higher-valency vaccine are presented in Table 3a and Table 3b, respectively.

Comparing between the PCV20-related strategy and lower-valency groups, the majority of analyses (and all of the three studies concentrated on European areas) found that PCV20 as a single-use vaccine significantly lowered the total costs and improved the QALY, leading to a negative ICER, suggesting that it was a highly cost-saving choice in comparison to PPV23 in individuals over the age of 18 [47,56,57]. In addition, PCV20 was dominant over PCV13 in the elderly (>65) [61]. The combined option PCV20/PPV23 still dominated over PPV23, but the medical cost reduction was five times greater (the most driven cost) when choosing PCV20 as a single-use vaccine, indicating that PCV20 could save greater costs than PCV20/PPV23 in adults > 18 in Denmark [57]. A similar pattern was recognized in England, where PCV20 offered a substantial decrease in both vaccine cost and direct cost but less QALY improvement compared to the combined option, demonstrating a superior money-saving choice [47]. In Italy, PCV20 or PCV20/PPV23 are associated with a lower cost and a higher QALY gain than PCV13/PPV23 in the all-risk group (including people over 50–100 who are immunocompetent, elderly people >65, and the immunocompromised group) [55]. However, the benefit of PCV20-related use was contrasted in the USA, where it was reported that either a single use or combined use of PCV20 did not prove to be a cost-effective choice over PPV23 for individuals over 65, with or without comorbidities, given that both of the ICERs far exceeded the WTP = USD 100,000–150,000 [48].

When comparing between PCV15-related programs and lower-valency groups, the economic benefit was documented only in combined use, by one analysis in Italy. PCV15/PPV23 was a dominant strategy over PCV13/PPV23 in adult 50–100 at risk at developing chronic conditions. However, PCV15/PPV23 demonstrated the highest cost reduction in immunocompromised patients (>18 years old), showing that vaccination is highly cost-effective in patients suffering from immune deficiency [55].

When assessing between a PCV20-related strategy and PCV15-related vaccination, the economic benefit was clearly superior in PCV20-related groups, irrespective of the age (18–64 or >65 years) and medical conditions. Compared to no vaccination, a single use of PCV20 required a higher vaccine cost but it was offset by the significant reduction in hospital expense, resulting in a lower total cost and a higher QALY value compared to PCV15. Hence, within the WTP, PCV20 was a more cost-effective and cost-saving choice than PCV15 in Europe and the USA [5,54,61]. Additionally, a significant portion of the analysis (6/11 studies) focused on comparing PCV20 and PCV15/PPV23 including in England, Belgium, Spain, Greece, Italy [5,47,55,59,60], and the USA [51]. While the USA suggested that PCV20 was a cost-effective option, with ICER being 9051 USD/QALY gained for elderly people >65 years of age [51], the remaining European nations revealed that PCV20 was dominant over PCV15/PPV23 by lowering the total cost and enhancing QALY for both adult and older people (with or without comorbidity). Even though PCV20 is simpler in terms of dose usage than combined PCV15/PPV23 (one dose vs. two doses), PCV20 still had a lower total vaccine price and a stronger effect in declining medical expenses, which ultimately contributed to the negative ICER. Furthermore, combining PCV20 with PPV23 was observed to save greater costs and result in a better QALY than PCV15/PPV23 in the immunocompetent group (at risk for chronic conditions, 50–100 years), according to one analysis in Italy [55].

**Table 3 healthcare-12-02490-t003:** (a) Incremental cost and health outcomes in higher-valency pneumococcal vaccines. (b) ICERs and sensitivity analysis outcomes of higher-valency pneumococcal vaccines.

**(a)**								
**Country, Currency, Ref**	**Intervention**	**Age**	**Vaccine Cost**	**Medical Cost**	**Indirect Cost**	**Total Cost**	**QALY**	**LY**
**All risk ***								
Denmark, 2022, EUR [57]	PCV20/PPV23 vs. PPV23 PCV20 vs. PPV23	≥18 high + ≥65	31,748,049 91,630,419	−57,782,573 −326,885,281	0	−53,766,066 −396,115,884	1350 5821	1433 5821
England, 2019, GBP [47]	PCV20 vs. PCV15/PPV23 PCV20 vs. PCV20/PPV23 PCV20 vs. PPV23	≥18 high + ≥65	−177.3 −236.5 378.70	1.2 −171.4 −538.31	- - -	−113.4 −235.3 −159,610,000	30,302 −343 91,375	- - -
Belgium, 2023, EUR [59]	PCV20 vs. PCV15/PPV23 PCV20 vs. No vaccination	≥18 high +≥65	−17,593,091 27,194,533	−9,314,716 −22,071,930	- -	−26,907,807 5,122,603	0.00016 0.00038	0.00020 0.00038
Spain, 2018, EUR [60]	PCV20 vs. PCV15/PPV23	≥60	−21,200,000	−64,600,000	-	−85,700	5870	8907
Italy, 2022, EUR [61]	PCV20 vs. PCV13 PCV20 vs. PCV15	65–74	40,568,000 40,568,000	−48,032,000 −40,205,000	- -	−7,464,000 −364,000	4734.0 3984.7	6581.6 5536.7
USA, 2017, USD [51]	PCV20 vs. No vaccination PCV15/PPV23 vs. No vaccination	≥65	- -	- -	- -	151 83	0.00072 0.00011	- -
USA, 2021, EUR [54]	PCV20 vs. No vaccination x3 PPV23 vs. No vaccination PCV15 vs. No vaccination	≥65	16,620,000 17,750,000 15,300,000	−7,420,000 −5,170,000 −4,310,000	−390,000 −220,000 −230,000	8,710,000 12,290,000 10,710,000	963 662 559	- - -
Greece, 2022, EUR [5]	PCV20 vs. PCV15/PPV23 PCV20 vs. PCV15	≥18	-	-	-	−48,858 −11,183	1536 1594	1883 1962
**Immunocompetent**								
Norway, 2022, EUR [56]	PCV20 vs. PPV23	≥18	67,200,826	−140,808,171	-	−73,607,345	7966	7584
Belgium, 2023, EUR [59]	PCV20 vs. No vaccination	65–84	10,461,746	−12,204,474	-	−1,742,727	0.0007	0.0008
USA, 2017, USD [48]	PCV20 vs. PPV23 PCV20/PPV23 vs. PPV23	≥65			- -	60.08 82.67	0.00035 0.00003	- -
Italy, 2013, EUR [55]	PCV15/PPV23 vs. PCV13/PPV23 PCV15/PPV23 vs. PCV20/PPV23 PCV15/PPV23 vs. PCV20 PCV15/PPV23 vs. No vaccination	50–100	0 0 53,184,529 185,043,395	-	-	−11,630,171 58,642,975 92,033,528 56,669,841	1488 −7559 −5255 15,718	4414 −22,401 −14,493 44,783
**Immunocompromised**								
Italy, 2013, EUR [55]	PCV15/PPV23 vs. PCV13/PPV23	≥18	0	-	-	−19,967,763	2778	9279
**(b)**								
**Country, Currency, Ref**	**Intervention**	**Age**	**ICERs**	**The Most Impactful Parameter in DSA**	**PSA**			**Conclusion**
					**% CE**	**WTP**	**Scatter Plot Distribution**	
**All risk ***								
Denmark, 2022, EUR [57]	PPV23/PCV20 vs. PPV23 PCV20 vs. PPV23	≥18	−44,326 −68,054	Time horizon	- -	- -	100% SE 100% SE	PCV20 was a dominant strategy in both cases
England, 2019, GBP [47]	PCV20 vs. PCV15/PPV23 PCV20 vs. PCV20/ PPV23 PCV20 vs. PPV23	≥18 high + ≥65	Dominant −686,948 Dominant	Robust	- - 85% 99%	- - 20,000 30,000	- - Majority SE Majority NE	PCV20 was cost-saving compared to PPV23 in adults aged 65–99 years and adults aged 18–64 years with underlying conditions
Belgium, 2023, EUR [59]	PCV20 vs. PCV15 / PPV23 PCV20 vs. No vaccination	≥18 high + ≥65	Dominant 4164	Cost and incidence of inpatients all-cause NBP, VE	100% 100%	35,000	100% SE 87% NE, 13% SE	PCV20 was cost-effective compared to no vaccination and cost-saving compared to PCV15/PPV23
Spain, 2018, EUR [60]	PCV20 vs. PCV15/PPV23	≥60	−14,605	Robust	100%	25,000	100% SE	PCV20 was more cost-effective than PCV15/PPV23
Italy, 2022, EUR [61]	PCV20 vs. PCV13 PCV20 vs. PCV15	65–74	Dominant 91	Robust	90% 90%	5000 5000	-	PCV20 was dominant over PCV13 and more cost-effective than PCV15
USA, 2017, USD [51]	PCV20 vs. No vaccination PCV15/PPV23 vs. No vaccination	≥65	210,529 728,423	Robust	>50% 6%	190,000 200,000	-	Within the WTP range, only PCV20 was favorable in non-Black people
USA, 2021, EUR [54]	PCV20 vs. No vaccination x3 PPV23 vs. No vaccination PCV15 vs. No vaccination	≥65	9051 18,559 19,162	Vaccine price, the VE, vaccine waning rate, the proportion of pCAP	90% - -	20,000	-	PCV20 was reported to be the cost-effective strategy if PCV10 was continued in children
Greece, 2022, EUR [5]	PCV20 vs. PCV15/PPV23 PCV20 vs. PCV15	≥18	Dominant Dominant	Robust	100% 100%	34,000	-	PCV20 was a dominant vaccination strategy over PCV15 alone or followed by PPV23
**Immunocompetent**								
Norway, 2022, EUR [56]	PCV20 vs. PPV23	≥18	−9420	Inpatient cost of NBPP, PCV20 vaccine price, PCV20 VE in NBPP	-	-	100% SE	PCV20 was cost-effective compared to PPV23
Belgium, 2023, EUR [59]	PCV20 vs. No vaccination	65–84	Dominant	Robust	-	35,000	77% SE, 23% NE	PCV20 was cost-saving in 65–84-year-old adults with a chronic underlying condition
USA, 2017, USD [48]	PCV20 vs. PPV23 PCV20/PPV23 vs. PPV23	≥65	172,491 3,115,054	Vaccine cost, VE, PP risk in high-risk adults	16–34%	100,000–150,000	-	PCV20 and PCV20/PPV23 were not more favorable economically than PPV23
Italy, 2013, EUR [55]	PCV15/PPV23 vs. PCV13/PPV23 PCV15/PPV23 vs. PCV20/PPV23 PCV15/PPV23 vs. PCV20 PCV15 /PPV23 vs. No vaccination	50–100	Dominant Dominated Dominated 3605	Probability of AMR for NBPP, age-specific utility, serotype-specific vaccine efficacy in NBPP	0% Dominated Dominated 0%	40,000	SE 100% NW 100% NW 100% NE	Sequential vaccination with either PCV15 or PCV20 combined with PPSV23 led to better health outcomes than PCV13/PPV23 and no vaccination
**Immunocompromised**								
Italy, 2013, EUR [55]	PCV15/PPV23 vs. PCV13/PPV23	≥18	Dominant	Probability of AMR for NBPP, age-specific utility, serotype-specific vaccine efficacy in NBPP	Regardless	100%	100% SE	PCV15/ PPSV23 was cost-saving compared to PCV13/PPV23

Abbreviation: QALY: quality-adjusted life-year; LY: life-years; PCV13/PPV23: PCV13 followed by PPV23; PCV15/PPV23: PCV15 followed by PPV23; PCV20/PPV23: PCV20 followed by PPV23; x3 PPV23: three doses of PPV23. All risk *: including immunocompetent and immunocompromised groups. PCV13/PPV23: PCV13 followed by PPV23; PCV15/PPV23: PCV15 followed by PPV23; PCV20/PPV23: PCV20 followed by PPV23; x3 PPV23: three doses of PPV23; NBPP: non-bacteremic pneumococcal pneumonia; NBP: non-bacteremic pneumonia; pCAP: pneumococcal community-acquired-pneumonia VT: vaccine type; VE: vaccine effectiveness, AMR: antibiotic resistance, SE: South East, NE: North East, NW: North West; ICERs: incremental cost-effectiveness ratio; DSA: deterministic sensitivity analysis; PSA: probabilistic sensitivity analysis; WTP: willingness to pay. All risk *: including immunocompetent and immunocompromised groups.

The DSA result showed that the model parameters were robust to the ICER in most of the studies (6/11 articles) [5,47,51,59,60,61]. The PSA result demonstrated that the majority of simulations for PCV20 concentrated on the South-East quadrant [47,56,57,59,60] when compared to PPV23 and PCV15/PPV23. Most of the ICER values of PCV20 had a probability of being cost-effective of more than 85% based on the WTP [5,47,59,60,61], indicating that using PCV20 saved more costs and improved quality of life compared to PPV23, PCV13, PCV15, and PCV15/PPV23. In the immunocompromised population, PCV15/PPV23 was 100% cost-effective regardless of the WTP compared to PCV13/PPV23 [55].

## 4. Discussion

### 4.1. Lower-Valency Vaccines

Our findings demonstrate that administering PCV13/PPV23 in older adults ≥65 would be the most beneficial regarding net cost-saving and QALY gain [45,49,50], which differs from the 2019 ACIP recommendation. This guideline suggested that any older adult ≥65 was recommended to receive a single dose of PPV23 [63]. Our results also found that, in immunocompromised patients, PCV13/PPV23 would be highly cost-saving compared to PPV23 alone. The cost-saving potential of the PCV13/PPV23 sequence may be attributed to its better protection against pneumonia. Data illustrated that the efficacy against PP in PPV23, PCV13, and PCV13/PPV23 was −10% to 11%, 40–79%, and 39–83% [64]. Evidence shows that PPV23 is highly effective in preventing VT-IPD, with 45% protection in VT-IPD and 74% against IPD, but it exhibited relatively poor protection against all-cause pneumonia [64]. Following a systematic review in 2019 by Jacob et al., PPV23’s efficacy ranged from 3% to 16% against all-cause pneumonia and was more potent in younger ages [65]. At the same time, PCV13 provided robust protection against vaccine serotype invasive pneumococcal disease and vaccine serotype pneumococcal pneumonia [66]. Following estimation, PCV13 could provide 41 to 71% protection against vaccine-type CAP and 30.6% for overall PP [67]. In the long term, PCV13 can maintain effectiveness against VT-CAP for 4 years, whereas PPV efficacy wanes over time [64]. It was observed that from the first year to the fifth year, the effectiveness of PPV23 drops from 74% to 15%, necessitating revaccination every 5–10 years [33]. According to Asai et al.’s research, the order of immunizing PCV13 before PPV23 would promote memory B cells and trigger the immune system, which is more likely to better enhance the PPV’s immune response [68].

However, the use of PCV13/PPV23 remains controversial within immunocompetent groups. Wateska et al. (2020) found PPV23 to be more financially favorable than the sequential vaccination in adults aged 65 and older [46], whereas Wateska et al. (2019) reported contrary results for adults aged 50–64 [53]. Furthermore, PPV23 was revealed to be ineffective against ACP in immunocompetence, according to a systematic review by Diao et al. in 2016 [69]. According to Choi et al., the PCV13/PPV23 regimen should be taken into consideration for inclusion in the National Immunization Program (NIP) for all-risk individuals 65 years of age and above, as it was determined to be more budget-friendly than PPV23 alone [45].

In terms of single-use, compared to no vaccination, PCV13 helps to decrease cost and increase QALY more strongly than PPV23 in older people ≥60 or ≥65 in Asia (Japan, Thailand) and Portugal [41,42,62], in contrast to the Netherlands [52]. It can be noted that, in Europe, the recommendation for older people still varies. A review assessing the trend in recommendations across six countries with a long history of using PCV—including the USA, France, Germany, the Netherlands, Spain, and the UK—highlights these discrepancies [70]. Apart from the USA (which still recommends PCV13, although this is based on decision-making), the five subsequent European nations were against authorizing the use of PCV13 in the older populations (≥60 or ≥65) due to the imbalance between the potential disease reduction capability and the high price of vaccine management [70]. Given that the reduced IPD cases in adults mainly benefit from the herd effect in pediatric NIP but not from the direct effectiveness of PCV13 itself, including PPV23 as a routine vaccine program would be a cost-effective choice since it has broader coverage against IPD and a lower price [70]. Following a systematic study in Europe 2023 assessing the NIP recommendation for adults in 26 countries, it is estimated that the majority are currently using PCV13/ PPV23; the second most common recommendation is PPV23 alone [71].

### 4.2. Higher-Valency Vaccines

Regarding broader-serotype pneumococcal vaccines, our findings indicate that the PCV15-related strategy and PCV20-related strategy are more cost-effective than the lower-valency vaccine. It is estimated that 43–60% and 63–72% of the PD episodes were contributed to by the PCV15 serotype and PCV20 serotype after the PCV13 pediatric program, proving that both PCV15 and PCV20 are appropriate and reasonable choices to prevent PD, thus saving costs [64].

The data relating to the cost-effectiveness of the PCV15-related vaccine were relatively limited since only one article discussed it, indicating the superior cost-reducing effect of PCV15/PPV23 over PCV13/PPV23 for immunocompetent adults 50–100 years old, as well as immunocompromised group, in Italy [55]. In contrast, the financial advantage of the PCV20-related program was discussed in various countries. However, the recommendations varied between Europe and the USA. Compared to PPV23, a single dose of PCV20 or combined PCV20/PPV23 would be the dominant strategy for adults 18–64 and older people over 65 in Denmark, England, and Norway. In addition, PCV20 alone would be more cost-effective since it reduces direct costs more strongly than PCV20/PPV23. The rationale for the superior cost-reducing effect of PCV20 over PPV23 is based on the most prevalent serotype coverage and the decreased vaccine-waning [57]. In European countries, serotype replacement has become prevalent, with pneumococcal disease caused by the PCV20 serotype accounting for more than 60% of IPD adult cases during 2018–2019 [72]. Nevertheless, in the USA, it concluded that the higher-valency program (PCV20, PCV20/PPV23, PCV15, or PCV15/PPV23) was considerably less cost-effective than the current program of PPV23 for elderly people ≥65 since the probability of being cost-effective within a WTP of 150,000 USD/QALY gained was less than 50% (the cost was inflated to the year 2017) [48]. One reasonable explanation might be the difference in the serotype contribution that most commonly causes illness. In the USA in 2016–2017, about 64.7% of IPD cases in adults were caused by 9N, 11A, 15A, 22F, 23A, 33F, and 35B [48].

Also, compared to the other higher-valency vaccine, PCV15, or sequential PCV15/PPV23, PCV20 was associated with more significant cost reductions and improvements in QALY, thus establishing it as the dominant intervention in terms of ICER [5,47,51,54,55,59,60,61]. Although the initial investment in PCV20 vaccine management is substantial, it is offset by the medical cost reduction, leading to a considerable overall cost reduction. The primary source of cost reduction stems from decreased hospitalization and treatment costs associated with pneumonia. The most substantial benefit of the PCV vaccine is that it reduces the considerable costs of preventing pneumonia cases, which both PCV15 and PCV20 have successfully inherited. However, the fundamental role of the newest vaccine is to fight against IPD cases, and PCV20 seems to protect against VT-IPD better than PCV15. Studies have shown that the IPD rate caused by unique PCV15 serotypes is much less severe, with 6.6% and 2,5% caused by serotypes 22F and 33F in Europe in 2018 [73]. However, IPD cases caused by PCV20 unique serotypes (8, 10A, 11A, 12F, and 15B) make up 29.3% [73]. According to a systematic review in 2023 by Grant et al., the PCV20 serotype was attributed to the most preventable IPD cases in 33 high-income nations that conducted pediatric PCV NIP (Korea, Denmark, Norway, England…) [10]. Furthermore, PD cases caused by PCV20 were associated with higher severity and a higher rate of antibiotic resistance, resulting in more complicated treatment, especially when it comes to managing IPD in adults [74]. This indicated that the disease burden was mainly focused on the PCV20 serotype; therefore, PCV20 might be the most cost-effective choice for alleviating the illness. Notably, the increasing trend of PCV20 unique-serotype IPD was higher in nations that have been conducting PCV13 NIP in pediatrics for a certain length of time (>3 years) compared to the countries recently authorizing PCV13 NIP (<3 years) (39.2% vs. 33.6%), focusing on high-income countries [75].

### 4.3. Implications

To simplify, PCV20 is superior (in cost-saving) to other pneumococcal vaccines in terms of cost-effectiveness, including PPV23, PCV13, PCV15, and PCV15/PPV23. PCV20 significantly reduces treatment costs, including vaccine and direct and indirect medical costs for adults 18–64 and >65, irrespective of comorbidities. In addition, a single use of PCV20 would be more cost-effective than PCV20/PPV23. Our finding supports the latest ACIP 2023 recommendation: for older people ≥65 years and with underlying medical conditions, and individuals 18–64 with a vaccine-naïve history, a single dose of PCV20 is recommended, replacing the previous guideline of one dose of PPV23 [76]. If PCV15 is selected, it should be followed by a single dose of PPV23 after a minimum interval of one year [76]. This strategy has been recently applied in the USA, Canada, and Australia for adults with underlying risk (higher-valency PCV combined with PPV23) [71].

In countries with a stable use of PCV13 NIP, our research proposes that PCV20 should be included in the NIP for adults 18–64 and people over 65, as PCV20 has been proven to mitigate the disease burden at an affordable cost. Including the pneumococcal vaccine in NIP can contribute to substantial benefits, including lessening the overall economic burden of PD as the greatest benefit. It was recognized that the financial benefit of higher-valency vaccines prioritized IPD over pneumococcal pneumonia. Even though pneumonia is more prevalent, IPD contributes to a higher mortality rate and consumes tremendous treatment costs (due to prolonged hospital stays) [64,77]. Given that Europe and Western countries have higher rates of the elderly population—who are more likely to develop PD—PCV20 would prevent tremendous amounts of PD, thus saving healthcare costs (mainly direct costs) and improving quality of life. Another positive outcome is to enhance vaccine coverage to create community immunization, which helps to limit disease transmission and ultimately prevent outbreaks of pneumococcal disease. This aligns with global recommendations such as the WHO aimed at reducing the burden of vaccine-preventable disease [78]. Also, fewer infections would lead to less need to use antibiotics, which is partially advantageous in mitigating the risk of antibiotic resistance. For countries currently using PPV23 as a routine program for elders over 65 and those under 65 at risk for pneumococcal disease, PCV20 as a single dose would be a reasonable replacement compared to PCV15 or sequential PCV15 followed by PPV23. However, several factors should be considered before including PCV in NIP, such as the geographical serotype distribution. In the USA, serotype 8 accounted for less than 1% of IPD cases in the 2012–2013 period, while it is more prevalent in European countries like Wales, France, and Spain and contributed to 20% of IPD cases in England [75,79,80]. Regarding the CAP problem, it was reported that one-third of the pneumococcal cases in the USA did not include serotypes covered by PCV15 and PCV20 [81]. The isolation culture showed that the most prevalent serotype was in PCV21 (9.3%), while the PCV20 and PCV15 serotypes accounted for 6.7% and 5.8% of CAP cases [81].

Choosing the most efficacious and cost-effective pneumococcal vaccine can be confusing. As vaccine development continues to evolve, newer PCVs are designed to keep up with changes in serotype distribution. While PCV13 and PPV23 still protect older adults against pneumococcal infections, they are less effective against specific dangerous serotypes and do not cover all the prevalent ones currently in circulation. Specifically, PCV13 has poor efficacy against serotype 3, which remains a significant cause of severe pneumococcal disease in older adults [82,83]. The development of higher-valency vaccines addresses the need for broader coverage and the shortcomings of lower-valency vaccines in combating specific serotypes. For example, PCV15 generates a stronger immune response against serotype 3 than PCV13, thereby improving protection against this severe strain. Furthermore, PCV20 provides even broader protection by targeting most of the emergent circulating serotypes responsible for IPD cases and antibiotic resistance. Given these factors, policymakers should consider regional serotype patterns before introducing new vaccines.

### 4.4. Strengths and Limitations

These studies’ results may not fully reflect the actual economic benefits of the vaccine for other groups, such as adults with chronic diseases or immunodeficiency.

Our research has several strengths. It is the first systematic review to assess the cost-effectiveness of pneumococcal vaccines in adults since the introduction of PCV15 and PCV20. Studies that analyzed the cost-effectiveness of PCV were selected from the most recent period to ensure that they were up to date with advances in vaccine technology and the clinical efficacy of PCV. Another strength of our research is its focus on the most vulnerable population for pneumococcal disease, those aged 65 and older. Approximately 70% of the included economic evaluations (16 out of 23 studies) concentrated on older adults, with the 65 and older age group being the most studied (11 out of 16 analyses). This demographic is a crucial target for pneumococcal vaccination and is often prioritized in government vaccination recommendations. In contrast, studies focusing on adults aged 18 and older comprised only 30.44% (7 out of 23) of the research, primarily addressing adults with chronic conditions within the 18–64 age range. Although pneumococcal disease is generally less severe in this younger population, there is increasing evidence supporting the cost-effectiveness of vaccination for them. Expanding vaccination policies to include more adults, especially in countries with a high pneumococcal disease risk among young adults, could be crucial.

Regarding economic analysis distribution, it can be observed that more than 70% of the included studies focused on CEA (16/23), while 26% focused on CUA (6/23), and the remaining study applied both CEA and CUA (1/23). This distribution is expected and understandable, particularly in the context of vaccines. CEA is a more straightforward and essential approach for policymakers because it estimates the cost required to achieve a specific health outcome, preventing pneumococcal episodes. It allows policymakers to directly assess the benefits of the PCV by determining how much it costs to avoid pneumococcal cases. In contrast, CUA also calculates the cost but takes a broader perspective by considering the quality of life of individuals, which is quantified using QALY. CUA answers how much it costs to gain one year of a healthy life (one QALY). This type of analysis is often more applicable to chronic diseases where long-term health and quality-of-life improvements are a significant focus. Therefore, the predominance of CEA in the included studies can be seen as a strength, considering its emphasis on observable immediate health outcomes.

Furthermore, the studies were all conducted in European countries, where the burden of pneumococcal disease, especially seroprevalence, is more pronounced than in other countries. This can be a strong plus for the study, yet the potential restrictions should be acknowledged. Specifically, the findings may not be fully applicable to Asian countries. One significant discrepancy lies in vaccine coverage. Developed nations typically have stable resources, allowing them to conduct widespread vaccination campaigns and achieve high vaccine coverage. In contrast, Asia faces significant challenges in implementing PCV programs. These challenges include logistical issues such as inadequate cold-chain technology for vaccine storage and high transportation costs, making delivering each dose prohibitively expensive. Additionally, the differences in antibiotic resistance and serotype epidemiology between Europe and Asia are substantial, meaning the cost-effectiveness observed in European countries may not translate to Asian contexts. Moreover, the assumptions were inconsistent with reality, as the studies made assumptions about low effectiveness: about one-third (8/23 studies) of the selected studies assumed very low or even 0% effectiveness of PPV23 in preventing pneumonia [5,45,46,47,48,53,56,57,62], which could lead to significant variability in ICER results between economic evaluations. However, the study also faced the following limitations. In this study, our research only focused on direct effects and ignored indirect effects due to the lack of information from the selected studies. Hence, the overall benefit of the vaccine may not be fully captured. Additionally, the cost-effectiveness analysis primarily addressed the general population rather than specific groups such as individuals with chronic diseases or immunodeficiency. Therefore, the data for these groups may not accurately reflect the actual economic benefits of the vaccine.

As presented in the Results Section above, of the 23 studies selected in this study, 12 were funded by pharmaceutical companies (11 were funded by Pfizer [5,41,45,47,49,56,57,59,60,61,62], and 1 was funded by Merck [55]). The funding of studies by pharmaceutical companies may create conflicts of interest, which may bias the results of the study in favor of the product of the company that funded it. In addition, some studies lack detailed information about the method, especially regarding evaluating the effectiveness of the PPV23 vaccine. The lack of information on some criteria (Items 18, 19, 21, 25) in all articles indicates that research reporting needs to be improved to provide adequate information and increase transparency for research. The lack of transparency about the method may increase the risk of bias, making the results of these studies less reliable. In addition, the included studies mainly evaluated the direct effectiveness of the vaccine in reducing treatment costs and hospitalization rates, and they did not take into account the herd effect from childhood vaccination programs, which can significantly affect the actual cost-effectiveness of the vaccine for adults.

## 5. Conclusions

In conclusion, based on the analysis of 23 cost-effectiveness studies of pneumococcal vaccines in adults, it can be seen that the PCV20 vaccine covers the most common serotypes and reduces the decline in immunity over time, effectively reducing the disease burden. PCV20 is more cost-effective and provides an improved quality of life (QALY) for vaccinated individuals compared to other pneumococcal vaccines, including PPV23, PCV13, PCV15, and PCV15/PPV23. Therefore, there should be a transition from the current PPV23 vaccination program to single-dose PCV20 for adults aged 18–64 at high risk of pneumococcal disease and those aged 65 years and older, as recommended by the ACIP 2023.

## 6. Future Directions

In the future, investment in the research and development of new, longer-lasting pneumococcal vaccines with broader serotype coverage is warranted. The future of pneumococcal conjugate vaccines (PCVs) for older adults is promising. Advances in vaccine technology, such as mRNA-based vaccines, could lead to faster and more flexible vaccine production. Developing new vaccines could help to improve the effectiveness of pneumococcal disease prevention and reduce the disease burden in communities. Current research has focused primarily on adults over 65 years of age and adults at high risk for pneumococcal disease, so further research is needed to evaluate the cost-effectiveness of pneumococcal vaccines in specific risk groups, such as immunocompromised individuals and people with chronic diseases, to make appropriate vaccination recommendations.

## Figures and Tables

**Figure 1 healthcare-12-02490-f001:**
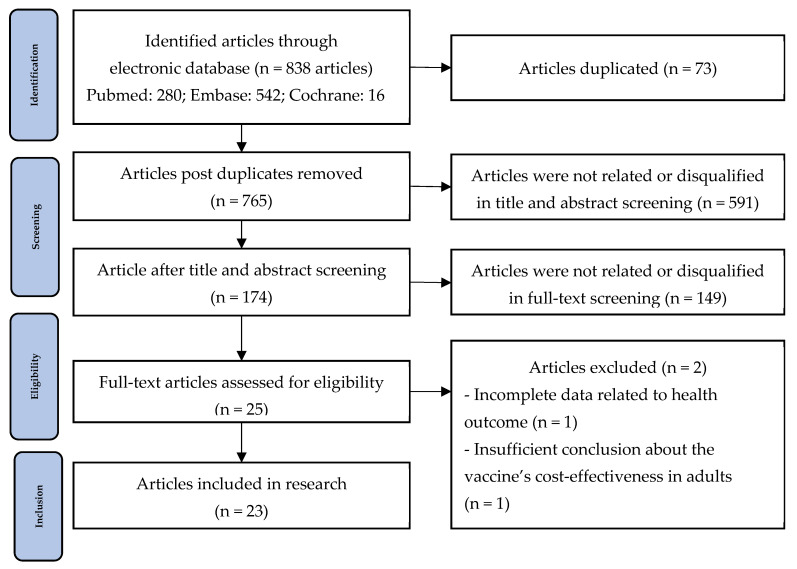
PRISMA flow diagram of the article selection process.

## Data Availability

Data were collected from three databases: PubMed, Embase, and Cochrane. These three databases are free, easy to access, and contain many studies, which facilitated this literature review.

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
