# Peer review of "Cost-Effectiveness of the Pneumococcal Vaccine in the Adult Population: A Systematic Review"

_healthcare, 2024, doi:10.3390/healthcare12232490_

Round 1

Reviewer 1 Report (Previous Reviewer 1)

Comments and Suggestions for Authors

Dear Authors,

Thank you for resubmitting your systematic review on the cost-effectiveness of pneumococcal vaccines in adults. The revised manuscript shows improvement, but several key issues still need to be addressed:

Major concerns

  1. Structure and Flow

·         "Future Directions" should be repositioned before the conclusion

·         Consider streamlining redundant content throughout the manuscript

·         Improve paragraph transitions for better readability

  1. Methodological Clarity

·         Resolve numerical discrepancies in study counts (CEA/CUA totals)

·         Complete the age distribution analysis for all 23 studies

·         Address the limitation of data being primarily from developed nations

  1. Writing Quality

·         Define abbreviations (PCV, PPV) at first mention (specially, the abstract part)

·         Revise unclear sentences in the abstract (particularly lines 19-22)

·         Consider professional English editing

Your manuscript provides valuable insights for vaccination policy, and these revisions would strengthen its contribution to the field.

Comments on the Quality of English Language

The revised manuscript shows noticeable deterioration in English language quality compared to the previous version. Multiple grammatical errors, awkward sentence structures, and unclear expressions are observed, particularly in the newly modified sections.

While your research content is valuable, the current language issues may hinder proper communication of your findings.

I strongly recommend thorough proofreading and professional English editing before resubmission to ensure your important research is presented with the clarity it deserves.

Author Response

Thank you for your precious time to review our study. Please see the attachment.

Reviewer 2 Report (Previous Reviewer 2)

Comments and Suggestions for Authors

Dear Editor,

I appreciate the authors' efforts in addressing my previous comments and making significant changes to the manuscript. These revisions have notably enhanced the rigour of the paper, which I believe will contribute positively to its overall impact.

I would like to suggest that including a discussion on the different types of pneumococcal vaccines could provide valuable insights for clinical practice. This addition would further strengthen the practical relevance of the manuscript.

However, I must regrettably reiterate my concern about the manuscript's novelty. While this remains a critical limitation, I acknowledge that in the context of different geographic settings, the novelty issue could be less significant, especially if the study has local relevance and implications. Considering that this study was conducted in Vietnam, its contributions to the local healthcare system and policy may justify its importance.

In light of the authors' comprehensive revisions and the contextual significance of their findings, I believe the manuscript, in its current form, is suitable for publication.

All the best

Author Response

Thank you for your precious time to review our study. Please see the attachment.

Round 2

Reviewer 1 Report (Previous Reviewer 1)

Comments and Suggestions for Authors

I praise the authors for their diligent work on the manuscript. Your thoughtful revisions have significantly enhanced this important contribution to the field. The dedication and scholarly rigor evident in your systematic approach is admirable, and your comprehensive analysis provides valuable insights for healthcare decision-makers worldwide.

However, a few minor areas still need attention to further polish this excellent work:

1. Accuracy Check

- On page 6, line 211, there appears to be a numerical discrepancy in reporting "2 studies discussed those aged ≥ 60 years (11/16 analysis)." The fraction should be (2/16) to maintain mathematical consistency with the total number of studies discussed.

2. Reporting Standards

- The Data Availability Statement could be strengthened by providing more specific details about data access

- The IRB exemption rationale would benefit from clearer explanation

- The CHEERS 2022 scoring methodology could be described more thoroughly

These minor refinements will further enhance what is already a well-conducted and valuable systematic review.

Author Response

Thank you for your precious time to review again our study. Please see the attachment

This manuscript is a resubmission of an earlier submission. The following is a list of the peer review reports and author responses from that submission.

Round 1

Reviewer 1 Report

Comments and Suggestions for Authors

Thank you for your comprehensive review of the cost-effectiveness of pneumococcal vaccines in adults. Your analysis provides important insights into the potential benefits of newer vaccines, particularly PCV20. However, there are several areas where the manuscript could be improved:

  1. The introduction is overly long and technical. Consider condensing the first three paragraphs and providing clear definitions for all acronyms (e.g., PP, ACP, VT-CAP) upon first use.
  2. The methods section, while generally well-structured, could benefit from clearer subheadings and a justification for the relatively short study period (2018-2024).
  3. Table 2 is extremely dense and difficult to read. Consider breaking it into multiple tables or using a different format to present this information more clearly.
  4. The results section contains some inconsistencies (e.g., mentioning 24 evaluations when there are 23 studies). Please review and correct these discrepancies.
  5. The discussion section, while comprehensive, could be more concisely structured to highlight key findings and implications.
  6. Throughout the paper, consider reducing the use of technical jargon and providing more explanations for a broader audience.

Addressing these issues would significantly enhance the clarity and impact of your valuable work. I look forward to seeing a revised version of this important contribution to the field.

Author Response

Please see the attachment. Thank you for taking the time to review our research.

Reviewer 2 Report

Comments and Suggestions for Authors

Dear Authors,

I appreciate the opportunity to review the manuscript titled "Cost-effectiveness of the Pneumococcal Vaccine in the Adult Population: A Systematic Review." The subject matter is undoubtedly important; however, after a thorough evaluation, I regret to inform you that the manuscript does not meet the necessary standards for publication in its current form due to concerns regarding originality, methodology, and the conclusions drawn.

Firstly, the topic of pneumococcal vaccine cost-effectiveness has been extensively researched, and the current manuscript lacks a novel perspective or significant contribution to the existing body of literature. A clear differentiation of this work from the numerous other studies on the subject is essential for it to stand out and provide added value to the field.

The abstract of the manuscript needs to deliver a more explicit 'take-home message' that succinctly conveys the essence and implications of the study's findings. A well-crafted abstract is crucial for engaging the reader and highlighting the significance of the research.

In the introduction section, a more focused approach is necessary. It should be reorganised to align strictly with the study's primary aim, clearly delineating how this work differs from the existing literature and its unique contributions.

Given that the effectiveness and economic benefits of pneumococcal vaccines have been well established, the manuscript would benefit from a deeper exploration of the economic advantages of vaccination programmes. This could enhance the relevance and applicability of the study’s findings.

Considering that many of the included studies overlap with the pandemic period, it is important to acknowledge the complexities introduced by concurrent respiratory illnesses during this time. I recommend a thorough evaluation of potential biases within the included studies, and a revision of the inclusion and exclusion criteria to address these concerns.

Table 2, in its current format, is not easily readable and requires reorganisation. Including a clear and legible table that displays the scoring results of the included studies, or at least providing it as supplementary material, would greatly improve the manuscript's comprehensibility.

Moreover, the manuscript suggests that some vaccines are more effective than others. However, it lacks a discussion regarding the specific vaccines used and potential conflicts of interest. Including this information is crucial for ensuring the transparency and reliability of the findings.

Lastly, the recommendations presented by the authors do not appear to differ significantly from current clinical practices. Thus, the study does not seem to offer substantial new insights or contributions to the literature.

In conclusion, while the topic is relevant, the manuscript requires significant revisions to address the aforementioned concerns. Due to the current limitations in originality, methodology, and the lack of substantial contributions to the field, I am unable to recommend this manuscript for publication.

Thank you for the opportunity to review this manuscript. I hope that the feedback provided will be of assistance to the authors in refining their work.

Yours sincerely,

Author Response

(The authors gave the same response as above.)

Reviewer 3 Report

Comments and Suggestions for Authors

Dear Authors,

It is interesting topic. 

1. Introduction

- all aspects connected to pneumococcal disease are presented; this disease is explained as well as their economic impact, however applying to 2017, so you could use some fresh data as in the meantime it could change. 

- different types of vaccinations are presented showing the possibility of different vaccination strategies; 

- the aim of the article is also presented (lines 80-83): 

" we aim to conduct a systematic review of cost-effectiveness studies in adults from the past six years (2018 to present) to highlight the economic benefits of the current programs (PCV13 and PPV23) and higher- valent vaccinations (PCV15 and PCV20)"

2. Method is well described:

- the research is based on systematic review by following the PRISMA 2020 checklist; and it addition, the methodological quality of all included articles was also conducted by using the the 28-item Consolidated Health Economic Evaluation Reporting Standards 2022 157 (CHEERS 2022) checklist.

- the time range is provided and it is very fresh 2018-2024

- also, all database is given and research is based on the wide range of data; 

- both process of selection and data extraction are very well explained / described; 

3. Results are presented in very systematic way - they starts with the presentation of selection process graphs then articles are characterized. 

4. Discussion - the results are well discussed; 

5. Conclusion and future direction of research are also indicated. 

I would say that good job. 

Author Response

Thank you for taking the time to review our research.

Round 2

Reviewer 2 Report

Comments and Suggestions for Authors

Dear Editor,

Despite the revisions made, I believe that the manuscript titled 'Cost-effectiveness of the pneumococcal vaccine in the adult population: A systematic review' still does not meet the necessary standards. This response is limited to a single page and does not address many of the critical points highlighted in my previous review. Furthermore, the responses provided lack the expected depth and clarity needed to support the requested revisions.

In light of these observations, it appears that the substantial changes necessary to enhance the manuscript have not been implemented. Consequently, I am inclined to conclude that the manuscript, in its current form, does not meet the required level of scientific rigor for publication in Healthcare.

Best regards,
